# What determines the 'culture of silence'? Disclosing and reporting sexual harassment among university employees and students at a large Swedish public university

Per-Olof Östergren[1]*, Catarina Canivet[1], Ulrika Andersson[2], Anette Agardh[1]

**1** Division of Social Medicine and Global Health, Department of Clinical Sciences, Lund University, Malmö, Sweden, **2** Department of Law, Lund University, Sweden

* per-olof.ostergren@med.lu.se

**Citation:** Östergren P-O, Canivet C, Andersson U, Agardh A (2025) What determines the 'culture of silence'? Disclosing and reporting sexual harassment among university employees and students at a large Swedish public university. PLoS ONE 20(3): e0319407. https://doi.org/10.1371/journal.pone.0319407

## Abstract

### Background

The MeToo-movement challenges the 'culture of silence' regarding sexual harassment (SH). There are few studies regarding this phenomenon in academic settings. The aim of this study was to investigate the relationship between having reported or disclosed SH, on the one hand, and background factors and general health and wellbeing of exposed individuals, types of SH and perpetrator characteristics, on the other hand.

### Methods and results

A questionnaire sent to all employees and students at a large Swedish university was returned by 33% (N = 2736) and 32% (N = 9677), respectively. This study is based on the 469 employees and 2044 students who affirmed that they had been exposed to SH at the university. Analyses were made by means of chi2 tests and logistic regression. Among employees, 38.8% had disclosed, i.e., talked to someone at the university about their experience, and 17.3% had formally reported, i.e., talked to someone at the university who had the obligation to act on this information. The corresponding figures among students were 11.2% and 4.0%. A higher professional rank was linked to lower disclosing and reporting behavior, although not statistically significantly. Among students, exposure to attempted or completed rape was linked to low rates of disclosing (24.3%) and reporting (8.1%). An asymmetrical power relationship was associated with higher rates of disclosing and reporting; although statistically significant for reporting only among employees, and for disclosing only among students. None of the health-related outcomes were related to disclosing or reporting.

### Conclusions

The study confirmed a culture of silence regarding SH in the university setting. Several factors were linked to this, which can be associated with gendered and other power

**Data availability statement:** The study is based on personal data for which legal restrictions for accessing the data apply. The data is deposited at Lund university, Sweden, and access is regulated by the Swedish Ethical Review Authority according to the terms under which ethical approval was granted. Data requests may be made to Lund University via registrator@lu.se.

**Funding:** This work was funded by the Swedish Research Council, grant number 2018-02457, main applicant AA. URL: https://www.vr.se/english.html The funders had no role in study design, data collection and analysis, decision to publish, or preparation of the manuscript.

**Competing interests:** The authors have declared that no competing interests exist.

relations in society at large and in the academic setting in particular. Similar factors affected employees as well as students, but the culture of silence seemed more pronounced among students.

## Introduction

Workplace sexual harassment (SH) is widely recognized as a health hazard with negative health outcomes, most importantly mental health problems [1–4]. Universities are workplaces also for students, for whom consequences of exposure to SH might be anxiety, increased alcohol use, post-traumatic stress disorder, physical pain, and impaired career opportunities [5].

One of the measures used by workplaces and institutions to handle the issue of SH is building reporting systems that lead to investigation and handling of complaints. However, there are barriers to such reporting, since offences in the sexual domain, compared to other offences, as a general rule tend to be massively underreported [1,5–7].

In research literature on SH at work it is acknowledged that the issue of the repercussions of formal reporting provokes reflection. An early study concerning individuals filing complaints about SH found that 65% of these involved job discharge – for the complainant [8]. Among field workers who had reported SH, only 18% were satisfied with the outcome of their reporting, according to one study [9]. A study from the military in the US showed that reporting did not improve – and in some cases worsened – job, psychological, and health outcomes for those who did [10]. 'Raising voice', defined as social support-seeking, confronting, or whistle-blowing, strategies used by victims of workplace mistreatment, including SH, was associated with several kinds of retaliation in one study among 1,167 public-sector employees [11]. Ostracizing targets of workplace sexual harassment has been described as punishment for those who disclose the abuse; however, performed not only by perpetrators. Also coworkers may use this mechanism, presumably as a way to distance themselves from those who 'make waves' by reporting [12]. Women's fear of being perceived negatively by their environment for reporting SH was seemingly confirmed as justified in one experimental study from 2019; participants were less likely to recommend a woman for promotion if she self-reported sexual harassment compared to other types of harassment or if the sexual harassment had been reported by a coworker [13]. Fear of covert retaliation was given as the reason by a junior female faculty member for not reporting a senior faculty member, who might in the future have an impact on her career in various subtle ways [14]. Finally, it is often stated that relying only on the reporting system for eliminating SH is ineffective, as in hunting for 'bad apples' one at a time, when instead one should look for organization-level solutions [15–16].

If formally underreporting sexual offences is the rule, talking to someone about what happened is, on the contrary, very common. This other person is typically a friend or a family member [17]. According to one recent meta-analysis, some recipient reactions to disclosures of sexual assault are not helpful, such as 'controlling', e.g., making decisions for the survivor or 'treating survivors differently', as in acting as if the survivor is 'damaged goods'; these were all associated with worse psychopathology [18]. However, positive reactions conveying validation or support may help a victim to heal and emotionally recover after a traumatic attack [19–20].

Notwithstanding all the mentioned barriers to reporting workplace SH, the MeToo movement in 2017 dramatically revealed the need to break the 'culture of silence' regarding SH in the workplace [21]. One of the most obvious reasons for reporting is that this may put an end to the harmful behavior. Another important motive, given by women who made the decision to report sexual assault to the police, was a desire to protect other women and girls [22]. One study investigated psychosocial well-being among 1,562 former military reservists

who had experienced SH during their service. Among those who did report, satisfaction with the reporting process and perceiving that the report had resulted in the harassment being addressed by authorities was associated with better post-harassment functioning and fewer symptoms of PTSD [23]. As stated in one article on organizational influences on victims' decision to report workplace sexual harassment, reporting with ensuing sanctions may deter others from engaging in similar behavior. Further, knowledge about SH and who is targeted, will enable organizations to make efforts to reduce negative consequences and to provide victims with needed support [24].

Increased insight into how reporting and informal disclosure of SH in university settings are related both to background factors of those offended and to circumstances related to the incidents will undoubtedly contribute to more knowledgeable strategies for prevention, diminishing, and handling of SH. Previous studies, mostly performed in the US, have shown that SH in university settings is formally underreported, but that many exposed individuals confide in someone, often a peer [25,26]. Women are exceedingly over-represented as victims of SH, but exposed men may be relatively less prone, both to report and to informally disclose [25,27,28]. Sexual offences of a more severe character, such as rape, will be reported more frequently than 'mere' SH [26,29]. Power relations, formal and informal, may be involved in victims' decision to disclose or not [30]. However, studies from other countries than the US are sparse, as well as studies with a larger representativity of both employees and students.

The aim of this study of employees and students at a large Swedish university was to investigate the relationship between having reported or disclosed SH that they had been exposed to and background factors of exposed individuals, types of SH, perpetrator characteristics, and exposed individuals' general health and wellbeing.

## Methods

### Design and study population

Details of the study setup and preliminary results have been presented elsewhere [31]. In short, within the framework of the 'Tellus' project at Lund University, Sweden [32], 33% of staff and PhD students (N = 2736) (hereafter labelled 'employees'), and 32% of students (N = 9667) at the university participated in a survey, which was sent by email in November 2019. The survey contained 120 items and focused on experiences of sexual harassment (SH), other derogatory treatment and discrimination, and on workplace-related health and psychological wellbeing. The present cross-sectional study involves participants who had affirmed exposure to SH, which was defined as having answered 'yes' to at least one of 10 questions describing various SH behaviors. They numbered 469 in the group of employees, i.e., 17.1% of participants, and 2044 in the group of students; 21.1% of participants. The set of 10 questions used to assess SH, the 'Lund University Sexual Harassment Inventory (LUSHI)' was recently validated, yielding two separate factors labeled 'unwanted sexual attention of soliciting type' and 'unwanted sexual attention of non-soliciting type' [33]. The *soliciting type of SH* represents explicit invitations to engage in a sexualized relationship, e.g., suggestions for dates, etc., while the *non-soliciting type of SH* behaviors reflects general attitudes which constitute a part of the psycho-social climate of a workplace. All included items are presented in Tables 3 and 4. See also the LUSHI instruments in the supporting information [S1_Table].

### Background variables

*Gender* was categorized with a question about gender identity as female, male or non-binary gender; however, when an answer to this question was lacking, the answer to a question about

gender assigned at birth was used. *Age* was categorized into groups, separately for employees and students. *Country of birth* was recorded as 'Sweden', 'Nordic countries (outside Sweden)', 'Europe (outside Nordic countries)', or 'outside Europe'. *Professional group among staff* was specified according to nine types, which were then aggregated into six categories, 'professor', 'senior lecturer', 'lecturer and researcher', 'PhD student', 'administrative and technical support staff', and 'others' on the basis of professional rank. *Employment form*; employees also answered one question about whether their employment was 'temporary' or 'permanent'. Students self-reported as *international student'* 'yes' or 'no'.

## Definitions of disclosure and reporting

All SH-exposed participants were asked about having talked about their SH experience with anyone at the university. For employees, the question was formulated as follows: 'Who at the university have you talked to about your experience? If the respondent affirmed any of the following answer alternatives, this was coded as *disclosure*: 'My supervisor or another person in a position of authority', 'Someone from human resources (HR)', 'Someone from my union', 'The occupational health services', 'A colleague', or 'Another person'; in the latter case, if the subsequent free-text precision clearly indicated a colleague or a person with a specified function at the university. Having made a formal report, *reported*, was defined as having affirmed the first two alternatives, i.e., talked to one's supervisor or another person in a position of authority or someone from HR.

All who refrained from answering this question were coded as having answered 'no' to both disclosure and reporting; i.e., we have no data on 'missing' for this variable regarding employees.

In the survey for students, the corresponding question was formulated as follows: 'Did you talk to anyone at the university about your experiences of this incident/these incidents? (Select all that apply)' with the following answer alternatives: 'Yes, an employee in a position of authority (course leader, director of studies, head of department etc.)', 'Yes, a study guidance counsellor', 'Yes, another person employed at Lund University', 'Yes, with my student union who helped me report this to the university', 'Yes, with the student ombudsman who helped me report this to the university', 'Yes, with my student union and/or the student ombudsman but asked them not to discuss this with the university', 'Yes, with a person engaged in the student life (for example a mentor, a student representative, foreman at a nation etc.)', 'Yes with student health', and 'No, I have not talked to anyone at the university about this'. Affirming any of the answer alternatives except the 'No'-answer was defined as having *disclosed*. Having affirmed any of the first five alternatives was defined as having *reported*.

Thus, in both data sets, anyone having reported was consequentially also coded as having disclosed.

## Definition of 'affirmation'

The question about disclosure or reporting appeared once in the survey, and its answer was thus linked to the individual, and not to any particular SH behavior or perpetrator characteristic. Since participants could affirm several SH behaviors as well as perpetrator characteristics, and in order to assess whether any of these were related to disclosure or reporting, the variable *affirmation* was used in the analyses. The number of affirmations thus differs from the number of SH-exposed participants.

## Perpetrators of SH

*Gender and function of perpetrator/perpetrators* were established, as well as *relationship of power* ('dominant/upper' or 'dependent/lower' or 'other') between perpetrator and respondent.

### Measures of health and well-being

*Self-rated health* [34] was assessed with the question 'How would you rate your health status in general?' with the answer alternatives 'very good', 'good', 'somewhat good', 'bad', and 'very bad', with dichotomization for the analyses between 'good' and 'somewhat good'. *Mental health* was assessed by two measures, firstly the 12-item version of the General Health Questionnaire (GHQ-12), with the 0-0-1-1 scoring method (range 0–12), and with poor mental health defined as a score of 3 or higher in the population of employees, and 4 or higher in the population of students [35, 36]. The Mental Health subscale of the 36-Item Short Form Health Survey questionnaire (SF-36) [37] was also used, with thresholds for poor mental health at the population tertile level, which was ≤60 for employees and ≤50 for students. As a measure of fatigue, we defined lack of fatigue as *vitality* using the Vitality score of the SF36 questionnaire [37], likewise with dichotomization at the tertile; ≤40 for employees and ≤30 for students. Finally, *ability to work or to study* was measured with a single question asking respondents to rank their current work ability/study ability on a 10-point scale [38], again with dichotomization at the population tertile.

### Statistical methods

The results are presented as numbers and frequencies of individuals with chi2 statistics for group differences, as well as odds ratios with 95% confidence intervals. Regarding the affirmations of SH behavior, perpetrator gender, perpetrator function, and perpetrator power position, these were added, yielding sums of number of affirmations. The percentage of affirmations in each category (SH behavior, perpetrator gender etc.) that was linked to individuals who had disclosed or reported was established. Thereafter, the associations between disclosure and reporting and each item in the category SH behavior, e.g., 'unwelcome comments', in the category SH behavior, were likewise established. Next, chi2 analyses were performed, thus assessing whether any particular SH behavior or perpetrator characteristic was more or less related to disclosure or reporting than the category as a whole. All analyses were performed using the IBM SPSS package, version 25. Significance was accepted at $p < 0.05$.

### Ethics approval and consent to participate

The study was approved by the Research Ethics Committee of Lund University (Dnr 2018/350). All methods were carried out in accordance with relevant guidelines and regulations. The data was collected via an invitation sent out to all employees and students at Lund University via e-mail and information was given in the letter, that participation was voluntary and that data would remain confidential and anonymous. The participants consented to participation by clicking on a link leading to the questionnaire, which was submitted anonymously after completion. The link was available between November 13, 2019 until February 3, 2020. This makes any further identification or communication with the participants impossible, i.e., the data set is indefinitely anonymized.

## Results

### Background factors and disclosing/reporting

Among employees, 38.8% disclosed this to another individual at the university and 17.3% made a formal report to their employer (Table 1). There were no gender-related differences in reporting frequency (also when non-binary individuals were excluded from the analyses). Persons 40 years and younger reported SH to a higher degree than those in older age groups (chi2 for trend; $p = 0.04$), with the same tendency regarding tendency to disclosure ($p = 0.08$). Individuals born outside Europe both disclosed and reported to a higher degree than those

**Table 1. Background characteristics of university staff and PhD students who had been exposed to sexual harassment (SH), and numbers and frequencies of those who disclosed their experiences to someone at the university and those who reported them formally to the university. Disclosure and reporting frequency differences between categories are presented with chi2 statistics and age-adjusted odds ratios (OR) with 95% confidence intervals (95% CI). By definition, all who reported also disclosed. N = 469.**

| | | All | Disclosed (N = 182) | | | | | Reported (N = 81) | | | | |
|---|---|---|---|---|---|---|---|---|---|---|---|---|
| | | | N | % | P for chi2 | OR | 95% CI | N | % | P for chi2 | OR | 95% CI |
| Gender | Women | 380 | 151 | 39.7 | | 1.0 | | 65 | 17.1 | | 1.0 | |
| | Men | 81 | 28 | 34.6 | | 0.84 | 0.51, 1.40 | 14 | 17.3 | | 1.10 | 0.58, 2.09 |
| | Non-binary | 8 | 3 | 37.5 | 0.69 | 0.87 | 0.20, 3.70 | 2 | 25.0 | 0.84 | 1.50 | 0.29, 7.68 |
| | Total | 469 | 182 | 38.8 | | | | 81 | 17.3 | | | |
| Age | ≤ 30 | 46 | 20 | 43.5 | | 1.78 | 0.78, 3.99 | 12 | 26.1 | | 1.62 | 0.63, 4.19 |
| | 31 – 40 | 106 | 51 | 48.1 | | 2.13 | 1.07, 4.22 | 24 | 22.6 | | 1.35 | 0.59, 3.06 |
| | 41 – 49 | 159 | 52 | 32.7 | | 1.12 | 0.58, 2.16 | 24 | 15.1 | | 0.82 | 0.36, 1.84 |
| | 50 – 59 | 102 | 42 | 41.2 | | 1.61 | 0.80, 3.21 | 11 | 10.8 | | 0.56 | 0.22, 1.41 |
| | ≥ 60 | 56 | 17 | 30.4 | 0.08* | 1.0 | | 10 | 17.9 | 0.04* | 1.0 | |
| Country of birth | Sweden | 377 | 146 | 38.7 | | 1.0 | | 63 | 16.7 | | 1.0 | |
| | Nordic country (outside Sweden) | 16 | 4 | 25.0 | | 0.47 | 0.15, 1.51 | 2 | 12.5 | | 0.61 | 0.13, 2.78 |
| | Europe (outside Nordic countries) | 41 | 13 | 31.7 | | 0.71 | 0.36, 1.42 | 6 | 14.6 | | 0.81 | 0.33, 2.02 |
| | Outside Europe | 35 | 19 | 54.3 | 0.13 | 1.80 | 0.90, 3.63 | 10 | 28.6 | 0.30 | 1.88 | 0.85, 4.13 |
| Professional position** | Professor | 51 | 18 | 35.3 | | 1.0 | | 4 | 7.8 | | 1.0 | |
| | Senior lecturer | 85 | 23 | 27.1 | | 0.64 | 0.30, 1.37 | 13 | 15.3 | | 1.81 | 0.55, 6.02 |
| | Lecturer and researcher | 72 | 30 | 41.7 | | 1.15 | 0.51, 2.58 | 13 | 18.1 | | 1.89 | 0.54, 6.66 |
| | PhD student | 53 | 24 | 45.3 | | 1.23 | 0.47, 3.17 | 12 | 22.6 | | 2.08 | 0.52, 8.29 |
| | Admin. and technical support staff | 192 | 78 | 40.6 | 0.09* | 1.15 | 0.56, 2.27 | 37 | 19.3 | 0.07* | 2.28 | 0.75, 6.99 |
| Employment form | Permanent | 363 | 134 | 36.9 | | 1.0 | | 58 | 16.0 | | 1.0 | |
| (missing = 6) | Temporary | 100 | 45 | 45.0 | 0.14 | 1.20 | 0.71, 2.03 | 23 | 23.0 | 0.10 | 1.22 | 0.64, 2.33 |

* chi2 for trend.

** Participants in the category "Others" (n = 16) were excluded from this analysis.

from other groups of countries, but the numbers were small and the differences found were not statistically significant. Professional position, a measure of hierarchy, seemed to be correlated with disclosing or reporting SH experiences, in that a higher rank was linked to lower disclosing and reporting behavior (p = 0.09 and 0.07, respectively). Individuals with temporary employment contracts disclosed and reported SH to a higher degree than those with permanent contracts, but the difference was not statistically significant.

A much lower proportion of the students exposed to SH had disclosed this to anyone at the university, and even fewer had made a formal report. The figures were 12.0 and 4.2%, respectively, for female students, and 7.9 and 2.6% for male students (Table 2). When excluding non-binary individuals from the analysis, the gender difference in frequency of disclosure became statistically significant; p = 0.02. However, the gender difference for reporting remained statistically insignificant.

Interestingly, age was not associated with the propensity to disclose the SH experience among the students in our study, while it was highly associated with reporting SH, i.e., older individuals reported their SH experience to a considerably higher degree. Country of birth was significantly associated with both disclosing and reporting SH; students born in Sweden were the ones with the lowest proportion of both behaviors, compared to the three other groups. Thus, among international students, disclosing and reporting were significantly more frequent than among national students.

**Table 2. Background characteristics of students who had been exposed to sexual harassment (SH), and numbers and frequencies of those who disclosed their experiences to someone at the university and of those who reported them formally to the university. Disclosure and reporting frequency differences between categories are presented with chi2 statistics and age-adjusted odds ratios (OR) with 95% confidence intervals (95% CI). By definition, all who reported also disclosed. N = 2044; missing answer to this question = 52.**

| | | All | Disclosed (N = 223) | | | | | Reported (N = 79) | | | | |
|---|---|---|---|---|---|---|---|---|---|---|---|---|
| | | | N | % | P for chi2 | OR | 95% CI | N | % | P for chi2 | OR | 95% CI |
| Gender | Women | 1590 | 191 | 12.0 | | 1 | | 67 | 4.2 | | 1 | |
| | Men | 382 | 30 | 7.9 | | 0.62 | 0.41, 0.92 | 10 | 2.6 | | 0.56 | 0.28, 1.10 |
| | Non-binary | 20 | 2 | 10.0 | | 0.79 | 0.18, 3.42 | 2 | 10.0 | | 1.99 | 0.43, 9.13 |
| | Total | 1992 | 223 | 11.2 | 0.07 | | | 79 | 4.0 | 0.14 | | |
| Age | 18 – 25 | 1681 | 187 | 11.1 | | 0.75 | 0.22, 2.58 | 57 | 3.4 | | 0.21 | 0.06, 0.74 |
| | 26 – 30 | 237 | 24 | 10.1 | | 0.68 | 0.19, 2.46 | 11 | 4.6 | | 0.29 | 0.08, 1.14 |
| | 31 – 40 | 53 | 9 | 17.0 | | 1.23 | 0.30, 5.06 | 8 | 15.1 | | 1.07 | 0.25, 4.48 |
| | ≥ 41 | 21 | 3 | 14.3 | 0.46* | 1 | | 3 | 14.3 | <0.001* | 1 | |
| Country of birth | Sweden | 1640 | 168 | 10.2 | | 1 | | 58 | 3.5 | | 1 | |
| | Nordic country (outside Sweden) | 40 | 10 | 25.0 | | 2.90 | 1.39, 6.05 | 4 | 10.0 | | 2.80 | 0.95, 8.22 |
| | Europe (outside Nordic countries) | 166 | 21 | 12.7 | | 1.26 | 0.78, 2.05 | 7 | 4.2 | | 1.12 | 0.50, 2.52 |
| (missing = 1) | Outside Europe | 145 | 24 | 16.6 | 0.003 | 1.72 | 1.08, 2.75 | 10 | 6.9 | 0.048 | 1.75 | 0.86, 3.54 |
| International student | Yes | 208 | 37 | 17.8 | | 1.84 | 1.25, 2.71 | 15 | 7.2 | | 1.91 | 1.06, 3.44 |
| | No | 1782 | 186 | 10.4 | 0.001 | 1 | | 64 | 3.6 | 0.01 | 1 | |
| (missing = 2) | | | | | | | | | | | | |

* chi2 for trend.

## Type of SH behavior and disclosing/reporting

When analyzing whether any particular type of SH behavior was associated with disclosing, there was no statistically significant relationship to be found among employees (Table 3). However, when the dichotomous scale was applied, it was clear that types of SH behaviors classified as 'soliciting', such as unwelcome gifts, pressuring for dates, unwelcome contacts online, by post or telephone, or stalking, were associated with a higher degree of disclosure than 'non-soliciting' behavior.

In the case of reporting, this tendency became even more evident, both when comparing all behaviors with each other and when applying the dichotomous scale (p = 0.001 and < 0.0001, respectively). As an example, unwelcome comments (i.e., non-soliciting SH) was associated with a reporting frequency of 21.1%, while those having received unwelcome gifts had reported in 34.8% of cases.

As seen in Table 3, the same pattern of associations between type of SH and tendency to disclose and report was evident among students, and in this group with strong statistical significance in all comparisons.

However, it is noteworthy that the SH item assessing sexual violence, i.e., attempted of completed rape, was associated with a lower rate of disclosure and reporting than several of the other types of SH. Among the 148 students having affirmed such an event, only 36, i.e., 24.3%, had disclosed to another person at the university that they had been exposed to any kind of SH, and only 12, i.e., 8.1%, had made a formal report.

## Perpetrator gender, function, and power position in relation to disclosing/reporting

Furthermore, we investigated whether perpetrator gender, function, or power position, as viewed by the respondent, was associated with disclosing and reporting. As seen in Table 5

**Table 3. Type of sexual harassment behavior in relation to disclosure and reporting, presented as numbers and percentages of affirmations, with chi2 statistics for disclosure/reporting frequency differences between categories. University staff & PhD students; N = 469, whereof 182 disclosed and 81 also reported.**

| | All affirmations | Disclosed | | | Reported | | |
|---|---|---|---|---|---|---|---|
| | | N of affirmations linked to participants who disclosed | % affirmations linked to disclosure | P for chi2 | N of affirmations linked to participants who reported | % affirmations linked to reporting | P for chi2 |
| *Non-soliciting SH:* | | | | | | | |
| Unwelcome comments | 304 | 127 | 41.8 | | 64 | 21.1 | |
| Unwelcome suggestive looks or gestures | 263 | 115 | 43.7 | | 57 | 21.7 | |
| Unwelcome 'inadvertent' brushing or touching | 157 | 64 | 40.8 | | 27 | 17.2 | |
| Unwelcome bodily contact such as grabbing or fondling | 99 | 38 | 38.4 | | 19 | 19.2 | |
| *Soliciting SH:* | | | | | | | |
| Unwelcome soliciting or pressuring for 'dates' | 121 | 66 | 54.5 | | 35 | 28.9 | |
| Unwelcome contact online, for example social media or email | 79 | 38 | 48.1 | | 23 | 29.1 | |
| Unwelcome contact by post or telephone | 72 | 42 | 58.3 | | 26 | 36.1 | |
| Unwelcome gifts | 46 | 24 | 52.2 | | 16 | 34.8 | |
| Stalking | 37 | 26 | 70.3 | | 19 | 51.4 | |
| Attempted or completed rape *(not included in the two SH scales)* | 10 | 6 | 60.0 | | 4 | 40.0 | |
| Sum of all affirmations | 1188 | 546 | 46.0 | 0.14 | 290 | 24.4 | 0.001 |
| All non-soliciting SH affirmations | 823 | 344 | 41.8 | | 167 | 20.3 | |
| All soliciting SH affirmations | 355 | 196 | 55.2 | 0.002 | 119 | 33.5 | <0.0001 |
| Sum of all affirmations | 1178 | 540 | 45.8 | | | | |

showing results for employees regarding perpetrator gender, differences were not statistically significant.

It mattered whether the individual who had been exposed to SH was in an unequal power position in relation to the perpetrator. In most such cases the perpetrator had the dominant position, but regardless of the direction of power the proportion who disclosed or reported any SH event was higher compared to exposure to SH events where the perpetrator and exposed individual were judged to have equal formal and informal power. However, this was only statistically significant regarding reporting.

Table 6 shows a different pattern with regard to students' experiences. Perpetrator female gender was related to a lower level of disclosing and reporting than perpetrator male, binary, or unknown gender. This pattern was statistically significant. As for the function of the perpetrator, the differences in the pattern of disclosing and reporting proportions were also statistically significant. Those who had been exposed by a PhD student or another university employee showed the highest proportions, 36.4 and 26.2%, respectively, for disclosure, and 25.0 and 19.7% for reporting.

Regarding the power relation between the exposed individual and the perpetrator, in similarity to the findings among employees, those who had perceived that there was an asymmetrical power relation showed higher proportions of disclosing or reporting. However, this was only statistically significant for disclosing

**Table 4. Type of sexual harassment behavior in relation to disclosure and reporting, presented as numbers and percentages of affirmations, with chi2 statistics for disclosure/reporting frequency differences between categories. Students; N = 2044; whereof 223 disclosed and 79 also reported; missing answer on this question = 52.**

| | | Disclosed | | | Reported | | |
|---|---|---|---|---|---|---|---|
| | All affirmations | N of affirmations linked to participants who disclosed | % affirmations linked to disclosure | P for chi2 | N of affirmations linked to participants who reported | % affirmations linked to reporting | P for chi2 |
| *'Non-soliciting' SH:* | | | | | | | |
| Unwelcome comments | 1025 | 141 | 13.8 | | 59 | 5.8 | |
| Unwelcome suggestive looks or gestures | 1182 | 150 | 12.7 | | 53 | 4.5 | |
| Unwelcome 'inadvertent' brushing or touching | 959 | 118 | 12.3 | | 32 | 3.3 | |
| Unwelcome bodily contact such as grabbing or fondling | 852 | 110 | 12.9 | | 30 | 3.5 | |
| *'Soliciting' SH:* | | | | | 0 | | |
| Unwelcome soliciting or pressuring for 'dates' | 572 | 77 | 13.5 | | 34 | 5.9 | |
| Unwelcome contact online, for example social media or email | 448 | 63 | 14.1 | | 31 | 6.9 | |
| Unwelcome contact by post or telephone | 152 | 36 | 23.7 | | 22 | 14.5 | |
| Unwelcome gifts | 77 | 25 | 32.5 | | 16 | 20.8 | |
| Stalking | 122 | 33 | 27.0 | | 19 | 15.6 | |
| Attempted or completed rape *(not included in the two SH scales)* | 148 | 36 | 24.3 | | 12 | 8.1 | |
| Sum of all affirmations | 5537 | 789 | 14.2 | <0.0001 | 308 | 5.6 | <0.0001 |
| All non-soliciting SH affirmations | 4018 | 519 | 12.9 | | 174 | 4.3 | |
| All soliciting SH affirmations | 1371 | 234 | 17.1 | | 122 | 8.9 | |
| Sum of all affirmations | 5389 | 753 | 14.0 | 0.0004 | 296 | | <0.0001 |

Since the analysis in Tables 3 and 4 was based on events of SH and one individual could report several events, we made a separate analysis of the association between perpetrator gender and tendency of disclosing/reporting SH, using individuals stratified by gender, and no statistical significant differences could be observed among women and men who had been exposed to SH by a person of opposite gender compared to same gender, neither among employees nor among students [S2_Table].

## Health and wellbeing in relation to disclosing/reporting

Finally, we investigated the association between self-rated general health, three measures of mental well-being, assessed by GHQ12 and the two SF-36 scales Mental Health and Vitality, and workability on the one hand, and the proportion of disclosure and reporting SH, on the other. Neither among employees (Table 7), nor among students (Table 8) were there any of the measured aspects related to a tendency to disclose or report SH.

## Discussion

This study aimed to shed light on the phenomenon called the 'culture of silence´ regarding sexual harassment at the workplace, in this case a large public university. The main objective of the MeToo-public media debate initiated in Sweden in 2017 was to target the culture of silence, and this in order to bring SH to the public attention, which in turn likely would lead

**Table 5. Perpetrator characteristics in relation to disclosure and reporting, presented as numbers and percentages of affirmations, with chi2 statistics for disclosure/reporting frequency differences between categories. University staff & PhD students; N=469, whereof 182 disclosed and 81 also reported.**

| | | Disclosed | | | Reported | | |
|---|---|---|---|---|---|---|---|
| | All affir-mations | N of affirmations linked to participants who disclosed | % affirmations linked to disclosure | P for chi2 | N of affirmations linked to participants who reported | % affirmations linked to reporting | P for chi2 |
| Gender of perpetrator | | | | | | | |
| Male | 377 | 154 | 40.8 | | 66 | 17.5 | |
| Female | 69 | 29 | 42.0 | | 17 | 24.6 | |
| Non-binary gender | 0 | 0 | 0 | | 0 | 0 | |
| Unknown gender | 24 | 7 | 29.2 | | 4 | 16.7 | |
| Sum of all affirmations | 470 | 190 | 40.4 | 0.81 | 87 | 18.5 | 0.69 |
| Function of perpetrator | | | | | | | |
| University employee | 320 | 122 | 38.1 | | 53 | 16.6 | |
| PhD student/research student | 47 | 27 | 57.4 | | 13 | 27.7 | |
| Student | 39 | 12 | 30.8 | | 7 | 17.9 | |
| Other person that the exposed person met through her/his/their work at the university | 93 | 35 | 37.6 | | 15 | 16.1 | |
| Sum of all affirmations | 499 | 196 | 39.3 | 0.15 | 88 | 17.6 | 0.57 |
| Power situation, if the perpetrator was a university employee | | | | | | | |
| The perpetrator had a dominant/upper position | 190 | 76 | 40.0 | | 40 | 21.1 | |
| The perpetrator had a dependent/lower position | 17 | 9 | 52.9 | | 4 | 23.5 | |
| Other person/relationship | 136 | 44 | 32.4 | | 11 | 8.1 | |
| Sum of all affirmations | 343 | 129 | 37.6 | 0.25 | 55 | 16.0 | 0.010 |

to the initiation of policy developments and effective interventions against SH [21]. There have been several studies which have examined the occurrence and analyzed probable causes of SH in different workplaces, including higher teaching institutions. However, few studies have focused on the factors linked to the propensity to disclose and report SH experiences – a core issue of the MeToo strategy – and this aspect is therefore in need of a systematic scrutiny.

Among the employees and students in our study who had affirmed in an on-line survey that they had been exposed to SH as defined by our 10-item instrument, only 38.8% of the employees and 11.2% of the students had 'disclosed', i.e., talked about their experience with another person at the university. 'Reporting' was defined as contacting an individual who formally is in a position where they should or could bring about a further investigation of any suspected case of SH, which violates university policy, or even common law. This contact could therefore lead to some form of disciplinary action and/or to a work or study environment intervention. As expected, the proportion who had reported their SH experience was even lower; 17.3% of employees and 4.0% of students had done so.

As outlined in the Introduction, underreporting of sexual offenses in general [6, 7], as well as when it comes to work-related SH [1], including SH in academia [5], could rather be described as 'the usual state of affairs'. Our results regarding formal reporting of SH thus confirm previous findings. As for 'disclosure' in our study, it must be noted that the question was formulated specifically as disclosure 'to someone at the university'. This will no doubt explain why the proportions of disclosure are lower than described in other studies, in which respondents to a high degree affirm having confided in friends or family members [26,29].

**Table 6. Perpetrator characteristics in relation to disclosure and reporting, presented as numbers and percentages of affirmations, with chi2 statistics for disclosure/reporting frequency differences between categories. Students; N = 2044, whereof 223 disclosed and 79 also reported; missing answer to this question = 52.**

| | All affirmations | Disclosed | | | Reported | | |
|---|---|---|---|---|---|---|---|
| | | N of affirmations linked to participants who disclosed | % affirmations linked to disclosure | P for chi2* | N of affirmations linked to participants who reported | % affirmations linked to reporting | P for chi2* |
| **Gender of perpetrator** | | | | | | | |
| Male | 1690 | 198 | 11.7 | | 68 | 4.0 | |
| Female | 341 | 27 | 7.9 | | 10 | 2.9 | |
| Non-binary gender | 11 | 6 | 54.5 | | 4 | 36.4 | |
| Unknown gender | 38 | 4 | 10.5 | | 2 | 5.3 | |
| Sum of all affirmations | 2080 | 235 | 11.3 | <0.0001 | 84 | 4.0 | <0.0001 |
| **Function of perpetrator** | | | | | | | |
| University employee | 122 | 32 | 26.2 | | 24 | 19.7 | |
| PhD student | 44 | 16 | 36.4 | | 11 | 25.0 | |
| Person that I met through my work placement | 109 | 15 | 13.8 | | 12 | 11.0 | |
| Person external to Lund Univ., that I met through my studies | 86 | 8 | 9.3 | | 4 | 4.7 | |
| Student | 1726 | 183 | 10.6 | | 46 | 2.7 | |
| Sum of all affirmations | 2087 | 254 | 12.2 | <0.0001 | 97 | 4.6 | <0.0001 |
| **Power situation, if the perpetrator was a university employee** | | | | | | | |
| Course director/Examiner or Dir. of studies/Program dir. | 61 | 20 | 32.8 | | 17 | 27.9 | |
| Other teacher or researcher | 57 | 12 | 21.1 | | 8 | 14.0 | |
| Administrative staff or other employee | 27 | 9 | 33.3 | | 6 | 22.2 | |
| Sum of all affirmations | 145 | 41 | 28.3 | 0.44 | 31 | 21.4 | 0.28 |
| **Power situation, if the perpetrator was a student** | | | | | | | |
| The perpetrator had a dominant/upper position | 311 | 65 | 20.9 | | 12 | 3.9 | |
| The perpetrator had a dependent/lower position | 102 | 19 | 18.6 | | 5 | 4.9 | |
| Other person/relationship | 1477 | 132 | 8.9 | | 36 | 2.4 | |
| Sum of all affirmations | 1890 | 216 | 11.4 | <0.0001 | 53 | 2.8 | 0.18 |

These results seem to convincingly corroborate the existence of a strong culture of silence, especially regarding the low proportion of individuals who actually report their SH experience. Reporting could, at least theoretically, lead to SH cases being formally managed by the university and also contribute to the statistics on this type of work- and study-related environment risk for health and well-being among employees and students. The very low figures found regarding students' reporting of SH, which were implied in a previous qualitative study from this university [32], could here be quantified and verified. Furthermore, the proportion among students who had at least disclosed their SH experience to another person at the university was, even considering the limitation described above, comparatively low. Thus, the culture of silence regarding SH appears to still be reigning in the general environment of the university, especially among students.

In many studies, men have been shown to underreport SH to a higher degree than women [25,27,28]. In our study, this was only the case for male students regarding disclosure; otherwise there were no statistically significant gender differences in disclosing or reporting. This

**Table 7.  Health and workability in relation to disclosure and reporting frequency. University staff and PhD students; N=469.**

|  |  | All | Disclosed (N = 182) | | | Reported (N = 81) | | |
|---|---|---|---|---|---|---|---|---|
|  |  |  | N | % | P for chi2 | N | % | P |
| Self-rated health *(missing 6)* | Very good or good | 325 | 126 | 38.8 |  | 55 | 16.9 |  |
|  | Somewhat good, bad, or very bad | 138 | 55 | 39.9 | 0.83 | 25 | 18.1 | 0.76 |
| Mental health, measured by GHQ *(missing 5)* | Good | 289 | 109 | 37.7 |  | 51 | 17.6 |  |
|  | Poor | 175 | 73 | 41.7 | 0.39 | 30 | 17.1 | 0.89 |
| Mental health, measured by SF36 *(missing 17)* | Good | 255 | 93 | 36.5 |  | 41 | 16.1 |  |
|  | Poor | 197 | 85 | 43.1 | 0.15 | 37 | 18.8 | 0.45 |
| Vitality, measured by SF36 *(missing 25)* | High | 274 | 100 | 36.5 |  | 48 | 17.5 |  |
|  | Low | 170 | 74 | 43.5 | 0.26 | 30 | 17.6 | 0.97 |
| Workability *(missing 9)* | Good | 327 | 132 | 40.4 |  | 57 | 17.4 |  |
|  | Poor | 133 | 48 | 36.1 | 0.39 | 23 | 17.3 | 0.97 |

**Table 8.  Health and ability to study in relation to disclosure and reporting frequency. Students. N=2044; missing answer on this question=52.**

|  |  | All | Disclosed (N = 223) | | | Reported (N = 79) | | |
|---|---|---|---|---|---|---|---|---|
|  |  |  | N | % | P for chi2 | N | % | P |
| Self-rated health *(missing 17)* | Very good or good | 1200 | 138 | 11.5 |  | 46 | 3.8 |  |
|  | Somewhat good, bad, or very bad | 775 | 82 | 10.6 | 0.53 | 32 | 4.1 | 0.74 |
| Mental health, measured by GHQ *(missing 14)* | Good | 943 | 99 | 10.5 |  | 38 | 4.0 |  |
|  | Poor | 1035 | 121 | 11.7 | 0.40 | 40 | 3.9 | 0.85 |
| Mental health, measured by SF36 *(missing 30)* | Good | 1207 | 133 | 11.0 |  | 44 | 3.6 |  |
|  | Poor | 755 | 84 | 11.1 | 0.94 | 33 | 4.4 | 0.42 |
| Vitality, measured by SF36 *(missing 36)* | High | 1327 | 145 | 10.9 |  | 53 | 4.0 |  |
|  | Low | 632 | 71 | 11.2 | 0.84 | 23 | 3.6 | 0.70 |
| Ability to study *(missing 44)* | Good | 1140 | 124 | 10.6 |  | 44 | 3.8 |  |
|  | Poor | 780 | 96 | 12.3 | 0.24 | 33 | 4.2 | 0.60 |

might be understood as a sign of better-than-average gender equality in our setting. However, this interpretation is probably too simplistic, also because men and women have different ways of thinking about and dealing with SH, which we have not been able to capture in our study [25,39,40].

We also noted that professional position among the employees was associated with the proportion who disclosed or reported their SH experience, such that individuals with higher professional rank were less prone to disclose or report. This might imply another mechanisms behind the culture of silence, such as self-blame because of not being able to defend oneself against derogatory behavior despite being in a high-status position. A high professional rank makes it more likely that the individual has had opportunities to observe the managerial level concerning the complicated and time-consuming procedures – often with non-conclusive outcomes – that tend to result from the reporting of SH in the academic setting.

It must also be noted that in our previous study, women in these groups, professors and senior lecturers, were sexually harassed to a greater degree than women in other professional groups[31]. The current findings are therefore bothersome, especially in the light of previous findings suggesting that SH of high-ranked women could be interpreted as a way of punishing women who violate traditional constructions of femininity by taking on positions of power in the workplace [41].

Both employees and students with a foreign background tended to disclose and report their experiences of SH to a higher extent, which could have several explanations. One could speculate that local students perceive themselves as more likely to become a permanent part of the relevant professional context, i.e., their teachers and course mates are individuals who theoretically could have a long-lasting influence over their career opportunities and could with some likelihood become future colleagues in mutual work- or research environments, especially in a relatively small country like Sweden. This might lead to more perceived negative effects of disclosing or making a formal complaint. In a similar vein, this could also explain why disclosing or reporting behavior was more common among individuals who were temporarily employed.

In this study we also investigated whether type of SH was associated with the proportion who disclosed or reported their SH experience. The instrument we used consists of ten different types of SH event, ranging from unwelcome looks to attempted or completed rape. We had found in a previous study [33], applying factor analysis, that five of the ten items could be grouped in a category labelled 'non-soliciting SH' and another four in a second, labelled 'soliciting SH'. The main difference between the factors was that SH in the latter group was directed more clearly towards a particular individual, seemingly aiming at creating a relationship. The item assessing attempted or completed rape fell outside both the mentioned categories [33]. Not surprisingly, we found a large variation in the proportion who disclosed or reported any SH experience depending on type. The tendency regarding both disclosing and reporting was higher among both employees and students concerning soliciting type of SH. This type of SH exposure might represent a more continuous experience and, since it is directed towards a particular individual, could be more threatening towards one's integrity. This might contribute to lower the threshold for disclosing and reporting it.

Previous studies have found higher reporting rates in cases with more severe victimization [25,26,29]. Therefore, the present finding of comparatively low rates of disclosure and reporting of attempted or completed rape – a criminal offence – was rather surprising and should perhaps be of particular concern. Among students, less than one of ten such instances was reported.

Also, in general, the proportion who disclosed or reported their SH experience was considerably lower among students compared with employees. This result seems to be in line with one previous study in which rates between students and faculty were compared; however, participants in that study were recruited through crowd-sourcing [30]. In the study by Kirkner et al., focusing on staff only, less than 10% had made a formal report, which thus is sizably lower than for employees in the present study. However, no data from undergraduate students in the same context are available [27].

The results quantify the much-discussed notion that the culture of silence regarding SH seems more prominent among students than among employees. There are several possible explanations for this. Firstly, the responsibility of the university is much less defined regarding the study environment of the students than regarding the work environment of the employees. Moreover, the majority of the SH events in the student group take place outside the university, in the student social or recreational arenas where there is a lack of formalized responsibility of the university. Despite this, there are often linkages between such student arenas and the proper study environment of the students. This could happen, for instance, when an individual is more and less forced to encounter a perpetrator at a course, or fears negative attitudes from course mates, without having clear options concerning how to deal with this. All this could contribute to the observed culture of silence regarding SH experiences noted among the students in our study.

It is difficult to understand why employees would tend to feel freer about reporting a female than a male perpetrator. In this context, it is important to remember that women are

overwhelmingly more often exposed to SH than men, and also that men are exposed to same-sex perpetrators much more often than women [1,31]. The 'common' perpetrator is thus a man, and one could therefore speculate that a woman acting in a traditionally 'un-feminine' way would more readily be identified as an aggressor than a man with a similar behavior.

Among employees, PhD students as perpetrators were likewise disclosed and reported relatively more often than other professional groups, though statistically not significant. Like other graduate students, PhD students can be seen as having an intermediary power position between students and faculty, whereby power and 'contrapower' mechanisms in this context may become particularly difficult to disentangle [42].

Among students, PhD students were likewise disclosed and reported at high rates, but this was more 'straightforward' in a power dynamics perspective, since this was the case also for other university employee perpetrators, in stark contrast to reporting rates for student perpetrators – and here the difference was statistically significant.

Finally, we assessed whether the explicitly perceived power relation to the perpetrator was associated with disclosing or reporting SH experiences. Both among employees and students, an asymmetrical power relationship – whether in a dominant or in a lower position with the perpetrator – was associated with higher rates of disclosure and reporting, albeit statistically significant for reporting only among employees and for disclosure only among students.

In summary, we interpret the results of our analyses as support for the hypothesis that power relations are important factors that shape the culture of silence regarding SH in a university context. The importance of gender could be understood in a larger context of gendered power relations in general as well as specifically in the academic context. A lower tendency among employees to disclose or report SH experiences if the perpetrator was a male, despite the fact that in most SH cases the perpetrator is of male gender and the exposed person of female gender, could be in line with the perception of being situated in a complex situation of power relations, which are a mix of both gendered power relations and formal and informal structural power relations specific to the academic world. The opposite pattern among students, a higher tendency to disclose or report SH if the perpetrator was of male gender, could perhaps be explained by the circumstance that the majority of perpetrators of SH in this group are other students and the most common arena for SH is student social life where it could be assumed that traditional male dominant behavior is less linked to other formal or informal power relations at the university, and therefore the barrier to disclosure and reporting is lower than among employees.

Nevertheless, our analysis showed at the same time that power relations indeed were important for disclosing or reporting SH among students. We do not interpret this as a contradiction, but rather this suggests that both the proposed mechanisms behind power relations relating to the culture of silence regarding SH exist in the academic setting and are overlayered. We also observed that not only being in a perceived subordinated power situation, but also being in a perceived superior position was linked to a higher tendency to disclose or report SH experiences. Due to small numbers, we could not further explore this finding and thus refrain from speculating about its possible explanations.

Finally, we wanted to investigate whether there is an association between disclosure and reporting of SH experiences on the one hand and self-rated health, three measures of mental wellbeing and workability/ability to perform one's studies on the other hand. We found that there were no associations between disclosing or reporting SH and any of the five mentioned outcomes, either among employees or students. There could be several explanations for this lack of associations in our study.

One possibility is the existence of a non-linear relationship between disclosure and reporting SH and these outcomes, e.g., a U-shaped relationship such that both very good or very bad

self-rated health or mental well-being could be associated with disclosure and reporting SH experiences. Also, there could exist a number of confounding and modifying factors, such as previous mental health status, work-place and other social support, which we are unable to analyze in this study due to insufficient numbers of cases. Unfortunately, our cross-sectional study design with a limited number of cases and comparatively short time window for exposure and outcome is not optimal for shedding light on the mentioned issues.

The knowledge gained from this study will be communicated to the university management and the local students' association through channels already built up in the conception phase of the study. The study has a clear participatory framework, where the mentioned stakeholders have taken part in choosing the study design, instruments and implementation mode and have had the opportunity to provide input to the interpretation of the findings [32]. This, we believe, will increase the likelihood of gained knowledge becoming translated into future policies and interventions and thus, hopefully, will circumvent previously reported difficulties in similar settings [43]. The final phase of the project will consist of evaluating the extent to which this was accomplished.

### Strengths and limitations

A major strength of our study is that it is based on a large target population, a university with a large staff and over 40,000 enrolled students. Few previous studies have addressed both employees and students, despite their shared work and study environment, which represents a common responsibility for the university management. Other strengths are that we used well-validated outcome measures. The overall response rate was 33% for employees and 32% for students. However, a comparison with register information from the university regarding basic demographic factors and employment status showed a high degree of agreement with our sample composition. Another limitation is the comparatively low number of cases when attempting to analyze detailed factors behind disclosing and reporting SH, despite the large primary target population. This limited the analytical approach in some of the analyses to test statistical significance as a trend, in case of variables at the ordinal level, or as a general pattern, in case of variables at the nominal level.

### Conclusion

Our study confirmed the existence of a considerable culture of silence regarding SH in the university setting and that several factors are linked to this. Furthermore, many of these can be linked to gendered and other power relations in society at large and in the academic setting in particular. Similar factors affected employees as well as students, but the culture of silence seems more pronounced among students, possibly due to the less clear role of the university regarding students' social life. This is of great concern since SH in this context could have severe implications for students' psychosocial study environment at the university. Knowledge acquired in this study can be useful for shaping adequate intervention programs as well when designing systems for reporting and managing cases of SH in the academic setting.

### Supporting information

**S1 Table.  Items of sexual harassment behavior in the Lund University Sexual Harassment Inventory (LUSHI).**
(DOCX)

**S2 Table.  Association between perpetrator gender and tendency of disclosing/reporting SH, using individuals stratified by gender.**
(DOCX)

## Author contributions

**Conceptualization:** Per-Olof Östergren, Catarina Canivet, Ulrika Andersson, Anette Agardh.

**Data curation:** Anette Agardh.

**Formal analysis:** Per-Olof Östergren, Catarina Canivet.

**Funding acquisition:** Per-Olof Östergren, Ulrika Andersson, Anette Agardh.

**Investigation:** Per-Olof Östergren, Anette Agardh.

**Methodology:** Per-Olof Östergren, Catarina Canivet.

**Supervision:** Per-Olof Östergren, Anette Agardh.

**Validation:** Per-Olof Östergren, Catarina Canivet.

**Writing – original draft:** Per-Olof Östergren, Catarina Canivet.

**Writing – review & editing:** Per-Olof Östergren, Catarina Canivet, Ulrika Andersson, Anette Agardh.

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
