## [Decision Letter · Decision Letter 0]

10 Sep 2024

PONE-D-24-08765What determines the ‘culture of silence’? Disclosing and reporting sexual harassment among university employees and students at a large Swedish public universityPLOS ONE

Dear Dr. Östergren,

Thank you for submitting your manuscript to PLOS ONE. After careful consideration, we feel that it has merit but does not fully meet PLOS ONE’s publication criteria as it currently stands. Therefore, we invite you to submit a revised version of the manuscript that addresses the points raised during the review process.

We look forward to receiving your revised manuscript.

Kind regards,

Michal Ptaszynski, PhD

Academic Editor

PLOS ONE

Journal Requirements:

3. In the online submission form, you indicated that data cannot be shared publicly because of the sensitive nature. Data are available from Lund University (contact via the correspondent author) for researchers who meet the criteria for access to confidential data.

Reviewers' comments:

Reviewer's Responses to Questions

**Comments to the Author**

1. Is the manuscript technically sound, and do the data support the conclusions?

Reviewer #1: Yes

Reviewer #2: Partly

Reviewer #3: Partly

2. Has the statistical analysis been performed appropriately and rigorously? 

Reviewer #1: Yes

Reviewer #2: Yes

Reviewer #3: Yes

3. Have the authors made all data underlying the findings in their manuscript fully available?

Reviewer #1: Yes

Reviewer #2: Yes

Reviewer #3: No

4. Is the manuscript presented in an intelligible fashion and written in standard English?

Reviewer #1: Yes

Reviewer #2: Yes

Reviewer #3: Yes

5. Review Comments to the Author

Reviewer #1: This is a very pertinent topic. I just wanted one addition by the authors that this study was carried out in an open society and still the reporting rate of SH is so low. What would be the situation in developing countries having very conservative enviourment. One or two references from 3rd world countires must be added in such an excellent study.

Reviewer #2: The paper addresses an important gap in the literature.

Introduction

In terms of the literature cited - some of it is quite dated so suggest you bring in newer references e.g. Systematic Review of Policies and Interventions to Prevent Sexual Harassment in the Workplace in Order to Prevent Depression - PMC (nih.gov)

Please explain what unwanted sexual attention of soliciting type’ and ‘unwanted sexual attention of non-soliciting type’ in the introduction - they are defined later, but bringing it in earlier would be helpful for the reader to better understand the study.

Data analysis and groupings

Agegroups you use are not inline with any global standard – under 30; 30-40 etc - Why was age 40 used as a cut off? e.g. your finding - Persons 40 years and younger reported SH to a higher degree than those in older age groups (chi2 for trend; p=0.04). Much of the literature on sexual assault on campus focus on first year students – can you pull out this data, as they are at thought to be at very high risk of assault and harassment.

Similarly you talk about 'older individuals' and that they report their SH experience to a considerably higher degree. Can you re-do the analysis with less ‘gross’ agegroups? And does that make any difference?

Word in table p15 - Inadvertent – would change – seems victim blaming

Tabel a3 – do you know if the victim was male or female?

Discussion

Add references to this section.

Not sure what this means - A high professional rank might also mean more experiences and knowledge about the perceived complications and often non-conclusive outcome of the university’s management of SH cases.

Comment on why local students do not disclose – more research needed.

Please gender your findings and the discussion especially regarding the victim versus the perpetrator. e.g. "We found that among employees, those who had been exposed to a female perpetrator were more likely to report this to the university, compared to other types of gender. However, the overall pattern concerning the association between perpetrator gender and disclosing or reporting SH experiences was not statistically significant among employees."

And here,

"The situation was strikingly different among students, where the proportion who disclosed or reported their SH experience was considerably lower among those who had been exposed by a female perpetrator, and this difference was statistically significant. Present perpetrator / victim data."

In terms of language – there is no use of the word victim / survivor – would try bring this language into the document - https://rainn.org/articles/key-terms-and-phrases#:~:text=We%20often%20use%20%E2%80%9Csurvivor%E2%80%9D%20to,to%20ask%20for%20their%20preference

Conclusion

Please detail how can this data be used, how are you the authors using this data, will they share it at the university? Clearly there is a need!

Reviewer #3: Review of PONE-D-24-08765

What determines the ‘culture of silence’? Disclosing and reporting sexual harassment among university employees and students at a large Swedish public university.

August 30, 2024

This manuscript describes a study of sexual harassment (defined broadly to include, for example, rape and attempted rape), disclosure, and reporting, as well as health outcomes among staff and students at a large Swedish university. The relative rates of incident acknowledgment, disclosure, and reporting were used to assess the degrees of a ‘culture of silence’ within both groups. The samples were relatively large for research on this topic, and the measures used had all been previously validated. The differences found within and between samples are potentially informative for the design of survivor support services. Please see more specific comments, below.

P6 Examples, or even the list, of the 10 items in the LUSHI would be useful for the reader of this manuscript, especially since they may not have access to Ostergren et al., 2022).

P7 Further rationale is needed for the decision to code missing data on disclosure and reporting as indicating a lack of both.

P9 Although the p-level for statistical significance was set at <.05, numerous findings throughout the Results section are reported as differing to various extents despite higher p-levels (on, e.g., P10). Descriptions of the findings would be clearer if the differences were characterized either as significant or not.

P23 The MeToo movement was actually started by Tarana Burke (e.g., Burke, 2021) in 2006.

P25 The phrase, “and not only because,” implies reasons in addition to the one given by the authors, but no others are given.

P26 “less than one of such instances” does not seem to make sense.

P27 One reason that employees might be more likely to report female than male perpetrators may be that they perceive a lower likelihood of retaliation from the former.

6. PLOS authors have the option to publish the peer review history of their article (what does this mean? ). If published, this will include your full peer review and any attached files.

**Do you want your identity to be public for this peer review?** For information about this choice, including consent withdrawal, please see our Privacy Policy .

Reviewer #1: **Yes: ** Dr. Tahir Jameel.

Reviewer #2: **Yes: ** Elizabeth Dartnall

Reviewer #3: **Yes: ** William F. Flack, Jr.

---

## [Author Response · Author response to Decision Letter 1]

14 Nov 2024

Review Comments to the Author

(Reviewer comments in plain text and responses in bold)

Reviewer #1: This is a very pertinent topic. I just wanted one addition by the authors that this study was carried out in an open society and still the reporting rate of SH is so low. What would be the situation in developing countries having very conservative enviourment. One or two references from 3rd world countires must be added in such an excellent study.

We thank the reviewer for this important comment. There is a well-known paradox, in that the reported prevalence of sexual harassment (SH) tends to be higher in societies which rank high in gender equity, whereas a lower prevalence is reported from settings with more conservative values concerning gender equity. One suggested explanation for this is that the value system affects what is acceptable, and thus in conservative environments the threshold seems to be higher for what is regarded as exposure to SH. This makes it difficult to compare the reporting and disclosing prevalences between studies with settings that differ in the described way. This is a very important discussion which deserves to be dealt with in its full complexity. However, given the limited space for discussion in the article format, we have abstained from making any comments about it in the article, especially since it is not directly related to the aim of the study. We have made a comment about the issue in a previous study, which concerned the prevalence of SH and its main determinants in our study population (Agardh et al., 2022).

Reviewer #2: The paper addresses an important gap in the literature.

Introduction

In terms of the literature cited - some of it is quite dated so suggest you bring in newer references e.g. Systematic Review of Policies and Interventions to Prevent Sexual Harassment in the Workplace in Order to Prevent Depression - PMC (nih.gov)

We are grateful for bringing the mentioned review to our attention. We have added this reference to new text in the discussion section, which addressed another comment below.

The knowledge gained from this study will be communicated to the university management and the local students’ association through channels already built up in the conception phase of the study. The study has a clear participatory framework, where the mentioned stakeholders have taken part in choosing the study design, instruments and implementation mode and have had the opportunity to provide input to the interpretation of the findings (Agardh et al., 2020). This, we believe, will increase the likelihood of gained knowledge becoming translated into future policies and interventions and thus, hopefully, will circumvent previously reported difficulties in similar settings (Diez-Canseco et al., 2022). The final phase of the project will consist of evaluating the extent to which this was accomplished.

Please explain what unwanted sexual attention of soliciting type’ and ‘unwanted sexual attention of non-soliciting type’ in the introduction - they are defined later, but bringing it in earlier would be helpful for the reader to better understand the study.

We are very grateful for this comment so that we can clarify the details of our definition of SH and the two mentioned subscales. We have added the text below in bold to do so.

The set of 10 questions used to assess SH, the ‘Lund University Sexual Harassment Inventory (LUSHI)’ was recently validated, yielding two separate factors labeled ‘unwanted sexual attention of soliciting type’ and ‘unwanted sexual attention of non-soliciting type’ (Ostergren, Canivet, Priebe, & Agardh, 2022). The soliciting type of SH represents explicit invitations to engage in a sexualized relationship, e.g., suggestions for dates, etc., while the non-soliciting type of SH behaviors reflects general attitudes which constitute a part of the psycho-social climate of a workplace. All included items are presented in Tables 2A and B. See also the LUSHI instruments in the supplementary material.

Data analysis and groupings

Agegroups you use are not inline with any global standard – under 30; 30-40 etc - Why was age 40 used as a cut off? e.g. your finding - Persons 40 years and younger reported SH to a higher degree than those in older age groups (chi2 for trend; p=0.04). Much of the literature on sexual assault on campus focus on first year students – can you pull out this data, as they are at thought to be at very high risk of assault and harassment.

Similarly you talk about 'older individuals' and that they report their SH experience to a considerably higher degree. Can you re-do the analysis with less ‘gross’ agegroups? And does that make any difference?

Here, we believe that the reviewer must have missed the fact that we use two separate age classifications for the two groups in our study, employees and students. Bearing that in mind, the presentation of the results will appear clearer and more logical. We include here an unformatted excerpt from the manuscript, showing firstly the results from employees with the corresponding text that was commented upon by the reviewer and thereafter the same procedure concerning students.

EMPLOYEES

disclosed reported

Age ≤ 30 46 20 43.5 1.78 0.78, 3.99 12 26.1 1.62 0.63, 4.19

31 – 40 106 51 48.1 2.13 1.07, 4.22 24 22.6 1.35 0.59, 3.06

41 – 49 159 52 32.7 1.12 0.58, 2.16 24 15.1 0.82 0.36, 1.84

50 – 59 102 42 41.2 1.61 0.80, 3.21 11 10.8 0.56 0.22, 1.41

≥ 60 56 17 30.4 0.08* 1.0 10 17.9 0.04* 1.0

Persons 40 years and younger reported SH to a higher degree than those in older age groups (chi2 for trend; p=0.04), with the same tendency regarding tendency to disclosure (p=0.08).

STUDENTS

disclosed reported

Age 18 – 25 1681 187 11.1 0.75 0.22, 2.58 57 3.4 0.21 0.06, 0.74

26 – 30 237 24 10.1 0.68 0.19, 2.46 11 4.6 0.29 0.08, 1.14

31 – 40 53 9 17.0 1.23 0.30, 5.06 8 15.1 1.07 0.25, 4.48

≥ 41 21 3 14.3 0.46* 1.0 3 14.3 <0.001* 1

Interestingly, age was not associated with the propensity to disclose the SH experience among the students in our study, while it was highly associated with reporting SH, i.e., older individuals reported their SH experience to a considerably higher degree.

Word in table p15 - Inadvertent – would change – seems victim blaming

We have carefully considered this comment. The word ’inadvertent’ was used in the survey instrument, however, with clear citation marks around this word. The citation marks were supposed to indicate that the respondent would have experienced the approach with brushing or touching as meant to appear non-intentional, as if happened by accident, while in reality it was performed on purpose. The same item appeared in the instrument used by Phillips et al. (Phillips et al., 2019).

Tabel a3 – do you know if the victim was male or female?

We would like to make the following clarification. As described in the Methods section, ’participants could affirm several SH behaviors as well as perpetrator characteristics’. In Tables 2A, 2B, 3A and 3B, data are presented regarding numbers of affirmations, which thus differ from the number of participants. Concerning perpetrator gender, there were fifteen in the group of employees and 87 in the group of students who affirmed having been exposed by two or more of the ‘perpetrator gender groups’ (i.e., men, women, non-binary, and unknown gender). In some of the analyses we separated groups of exposed persons by gender, and when doing so, as for instance regarding perpetrator gender and the case of female employees exposed, there was no statistically significant difference in the tendency to disclose depending on perpetrator gender. Given the complexity of these analyses and the comparatively small numbers involved, we thereafter chose to present data without separation by gender.

Discussion

Add references to this section.

We are grateful for the suggestion to add the major review (Diez-Canseco et al., 2022) article to the Discussion section, see previous response above.

Not sure what this means - A high professional rank might also mean more experiences and knowledge about the perceived complications and often non-conclusive outcome of the university’s management of SH cases.

We are grateful for this comment, since we realize that a clarification is called for. Please see the suggested re-formulation of the text in the manuscript, changes in bold.

A high professional rank makes it more likely that the individual has had opportunities to observe the managerial level concerning the complicated and time-consuming procedures – often with non-conclusive outcomes – that tend to result from the reporting of SH in the academic setting.

Comment on why local students do not disclose – more research needed.

We cannot draw any definite conclusions from our data concerning this. However, we are grateful for the comment, since we were able to clarify our thoughts on this matter. Please see the suggested change of text in the manuscript; changes are in bold.

One could speculate that local students perceive themselves as more likely to become a permanent part of the relevant professional context, i.e., their teachers and course mates are individuals who theoretically could have a long-lasting influence over their career opportunities and could with some likelihood become future colleagues in mutual work- or research environments, especially in a relatively small country like Sweden. This might lead to more perceived negative effects of disclosing or making a formal complaint. In a similar vein, this could also explain why disclosing or reporting behavior was more common among individuals who were temporarily employed.

Please gender your findings and the discussion especially regarding the victim versus the perpetrator. e.g. "We found that among employees, those who had been exposed to a female perpetrator were more likely to report this to the university, compared to other types of gender. However, the overall pattern concerning the association between perpetrator gender and disclosing or reporting SH experiences was not statistically significant among employees."

And here,

"The situation was strikingly different among students, where the proportion who disclosed or reported their SH experience was considerably lower among those who had been exposed by a female perpetrator, and this difference was statistically significant. Present perpetrator / victim data."

We are very grateful for this comment. We have changed the text in the Results section by deleting comments regarding results which were not statistically significant to the following:

Furthermore, we investigated whether perpetrator gender, function, or power position, as viewed by the respondent, was associated with disclosing and reporting. As seen in Table 3A showing results for employees regarding perpetrator gender, differences were not statistically significant.

In the Discussion section we omitted the following text.

We also analyzed whether certain characteristics of the perpetrator were associated with the proportion who disclosed or reported their SH experience. We found that among employees, those who had been exposed to a female perpetrator were more likely to report this to the university, compared to other types of gender. However, the overall pattern concerning the association between perpetrator gender and disclosing or reporting SH experiences was not statistically significant among employees. The situation was strikingly different among students, where the proportion who disclosed or reported their SH experience was considerably lower among those who had been exposed by a female perpetrator, and this difference was statistically significant.

In terms of language – there is no use of the word victim / survivor – would try bring this language into the document - https://rainn.org/articles/key-terms-and-phrases#:~:text=We%20often%20use%20%E2%80%9Csurvivor%E2%80%9D%20to,to%20ask%20for%20their%20preference

We are grateful for this comment which made us reflect more in-depth over our chosen terminology. We think that the observed choice stems from the tradition in epidemiology to avoid foreclosing the effect on the exposed individual in a study, in order not to be regarded as predetermining the outcome before making the analyses. In other words, if the individuals have been negatively affected, it should be evident from the analysis of data. We acknowledge that this seems to be very different in a judicial context, which logically is based on the already proven negative impact of a factor regulated, or in need of being regulated, by legislation, hence the term ‘victim’ seems most relevant. The term ‘survivor’ seems to imply an even more severe effect in the individual. Therefore, we prefer not to use the words “victim” or “survivor”.

Conclusion

Please detail how can this data be used, how are you the authors using this data, will they share it at the university? Clearly there is a need!

We are very grateful for this comment. It spurred us to add the following paragraph to the Conclusion section: (See also response to a previous comment from this reviewer.)

The knowledge gained from this study will be communicated to the university management and the local students’ association through channels already built up in the conception phase of the study. The study has a clear participatory framework, where the mentioned stakeholders have taken part in choosing the study design, instruments, and implementation mode, and have had the opportunity to provide input to the interpretation of the findings (Agardh et al., 2020). This, we believe, will increase the likelihood of gained knowledge becoming translated into future policies and interventions and thus, hopefully, will circumvent previously reported difficulties in similar settings (Diez-Canseco et al., 2022). The final phase of the project will consist of evaluating the extent to which this was accomplished.

Reviewer #3: Review of PONE-D-24-08765

What determines the ‘culture of silence’? Disclosing and reporting sexual harassment among university employees and students at a large Swedish public university.

August 30, 2024

This manuscript describes a study of sexual harassment (defined broadly to include, for example, rape and attempted rape), disclosure, and reporting, as well as health outcomes among staff and students at a large Swedish university. The relative rates of incident acknowledgment, disclosure, and reporting were used to assess the degrees of a ‘culture of silence’ within both groups. The samples were relatively large for research on this topic, and the measures used had all been previously validated. The differences found within and between samples are potentially informative for the design of survivor support services. Please see more specific comments, below.

P6 Examples, or even the list, of the 10 items in the LUSHI would be useful for the reader of this manuscript, especially since they may not have access to Ostergren et al., 2022).

As we responded to a comment from Reviewer #2, we are grateful for this observation and we have made a clarification as follows in the manuscript:

The soliciting type of SH represents explicit invitations to engage in a sexualized relationship, e.g., suggestions for dates, etc., while the non-soliciting type of SH reflects general attitudes which constitute a part of the psycho-social climate of a workplace. All included items are presented in Tables 2A and B. See also the LUSHI instruments in the supplementary material.

P7 Further rationale is needed for the decision to code missing data on disclosure and reporting as indicating a lack of both.

We are grateful for this comment, which pinpoints a weakness in a detail of the used questionnaire. We have revised the text accordingly. As we have now described more clearly in the manuscript, there was no possibility for the employees (albeit there was such an opportunity for students) to actively deny that they had talked to someone at the university about the offense that they had endured. We chose to code absence of choosing any of the presented

---

## [Decision Letter · Decision Letter 1]

18 Dec 2024

PONE-D-24-08765R1What determines the ‘culture of silence’? Disclosing and reporting sexual harassment among university employees and students at a large Swedish public universityPLOS ONE

Dear Dr. Östergren,

Thank you for submitting your manuscript to PLOS ONE. After careful consideration, we feel that it has merit but does not fully meet PLOS ONE’s publication criteria as it currently stands. Therefore, we invite you to submit a revised version of the manuscript that addresses the points raised during the review process.

We look forward to receiving your revised manuscript.

Kind regards,

Michal Ptaszynski, PhD

Academic Editor

PLOS ONE

Journal Requirements:

Reviewers' comments:

Reviewer's Responses to Questions

**Comments to the Author**

1. If the authors have adequately addressed your comments raised in a previous round of review and you feel that this manuscript is now acceptable for publication, you may indicate that here to bypass the “Comments to the Author” section, enter your conflict of interest statement in the “Confidential to Editor” section, and submit your "Accept" recommendation.

Reviewer #2: All comments have been addressed

Reviewer #3: All comments have been addressed

2. Is the manuscript technically sound, and do the data support the conclusions?

Reviewer #2: Partly

Reviewer #3: (No Response)

3. Has the statistical analysis been performed appropriately and rigorously? 

Reviewer #2: No

Reviewer #3: (No Response)

4. Have the authors made all data underlying the findings in their manuscript fully available?

Reviewer #2: No

Reviewer #3: (No Response)

5. Is the manuscript presented in an intelligible fashion and written in standard English?

Reviewer #2: Yes

Reviewer #3: (No Response)

6. Review Comments to the Author

Reviewer #2: I remain concerned about the decision to omit perpetrator data by gender, citing small sample sizes and complexity. While these challenges are understandable, it is ethically problematic to present perpetrator data without gendering it. Gender is central to understanding sexual harassment dynamics, and excluding it obscures important patterns and potentially reinforces biases.

Even if small numbers prevent robust statistical analysis, it remains vital to present the data transparently, with appropriate caveats about statistical limitations. I strongly recommend including the full gender breakdown of perpetrator data—men, women, non-binary, and unknown—while acknowledging any limitations. This will ensure ethical reporting and allow readers to critically interpret the findings without assumptions or gaps.

Reviewer #3: (No Response)

7. PLOS authors have the option to publish the peer review history of their article (what does this mean? ). If published, this will include your full peer review and any attached files.

**Do you want your identity to be public for this peer review?** For information about this choice, including consent withdrawal, please see our Privacy Policy .

Reviewer #2: **Yes: ** Elizabeth Dartnall

Reviewer #3: **Yes: ** William F Flack Jr

---

## [Author Response · Author response to Decision Letter 2]

25 Jan 2025

We are grateful for the comment, which made us add the requested analyses as ‘Supporting Information’ in addition to the following additin in the manuscript (2nd paragraph, page 22):

‘Since the analysis in Table 3 was based on events of SH and one individual could report several events, we made a separate analysis of the association between perpetrator gender and tendency of disclosing/reporting SH, using individuals stratified by gender, and no statistical significant differences could be observed among women and men who had been exposed to SH by a person of opposite gender compared to same gender, neither among employees nor among students (Supplementary material Tables 6 E-F).’

Yours sincerely,

Per-Olof Östergren, MD, PhD

First and corresponding author

---

## [Decision Letter · Decision Letter 2]

3 Feb 2025

What determines the ‘culture of silence’? Disclosing and reporting sexual harassment among university employees and students at a large Swedish public university

PONE-D-24-08765R2

Dear Dr. Östergren,

We’re pleased to inform you that your manuscript has been judged scientifically suitable for publication and will be formally accepted for publication once it meets all outstanding technical requirements.

Kind regards,

Michal Ptaszynski, PhD

Academic Editor

PLOS ONE

Additional Editor Comments (optional):

Reviewers' comments:

Reviewer's Responses to Questions

**Comments to the Author**

1. If the authors have adequately addressed your comments raised in a previous round of review and you feel that this manuscript is now acceptable for publication, you may indicate that here to bypass the “Comments to the Author” section, enter your conflict of interest statement in the “Confidential to Editor” section, and submit your "Accept" recommendation.

Reviewer #2: All comments have been addressed

2. Is the manuscript technically sound, and do the data support the conclusions?

Reviewer #2: Yes

3. Has the statistical analysis been performed appropriately and rigorously? 

Reviewer #2: Yes

4. Have the authors made all data underlying the findings in their manuscript fully available?

Reviewer #2: Yes

5. Is the manuscript presented in an intelligible fashion and written in standard English?

Reviewer #2: Yes

6. Review Comments to the Author

Reviewer #2: Thank you for addressing my comments - i have no further comments to make on this paper - i look forward to seeing it publishe

7. PLOS authors have the option to publish the peer review history of their article (what does this mean? ). If published, this will include your full peer review and any attached files.

**Do you want your identity to be public for this peer review?** For information about this choice, including consent withdrawal, please see our Privacy Policy .

Reviewer #2: **Yes: ** Elizabeth Dartnall

---

## [Editor Report · Acceptance letter]

PONE-D-24-08765R2

PLOS ONE

Dear Dr. Östergren,

I'm pleased to inform you that your manuscript has been deemed suitable for publication in PLOS ONE. Congratulations! Your manuscript is now being handed over to our production team.

Kind regards,

on behalf of

Dr. Michal Ptaszynski

Academic Editor

PLOS ONE